# Silver Nanoparticle-Embedded Conductive Hydrogels for Electrochemical Sensing of Hydroquinone

**DOI:** 10.3390/polym15112424

**Published:** 2023-05-23

**Authors:** Tingting Xu, Huanli Gao, Orlando J. Rojas, Hongqi Dai

**Affiliations:** 1Jiangsu Co-Innovation Center of Efficient Processing and Utilization of Forest Resources, International Innovation Center for Forest Chemicals and Materials, College of Light Industry and Food Engineering, Nanjing Forestry University, Nanjing 210037, China; 2Bioproducts Institute, Departments of Chemical and Biological Engineering, Chemistry and Wood Science, The University of British Columbia, 2360 East Mall, Vancouver, BC V6T 1Z3, Canada

**Keywords:** hydroquinone, silver nanoparticle-embedded, conductive hydrogel, electrochemical sensor

## Abstract

In this work, a conductive hydrogel was successfully synthesized, taking advantage of the high number density of active amino and hydroxyl groups in carboxymethyl chitosan and sodium carboxymethyl cellulose. These biopolymers were effectively coupled via hydrogen bonding with the nitrogen atoms of the heterocyclic rings of conductive polypyrrole. The inclusion of another biobased polymer, sodium lignosulfonate (LS), was effective to achieve highly efficient adsorption and in-situ reduction of silver ions, leading to silver nanoparticles that were embedded in the hydrogel network and used to further improve the electro-catalytic efficiency of the system. Doping of the system in the pre-gelled state led to hydrogels that could be easily attached to the electrodes. The as-prepared silver nanoparticle-embedded conductive hydrogel electrode exhibited excellent electro-catalytic activity towards hydroquinone (HQ) present in a buffer solution. At the optimum conditions, the oxidation current density peak of HQ was linear over the 0.1–100 μM concentration range, with a detection limit as low as 0.12 μM (signal-to-noise of 3). The relative standard deviation of the anodic peak current intensity was 1.37% for eight different electrodes. After one week of storage in a 0.1 M Tris-HCl buffer solution at 4 °C, the anodic peak current intensity was 93.4% of the initial current intensity. In addition, this sensor showed no interference activity, while the addition of 30 μM CC, RS, or 1 mM of different inorganic ions does not have a significant impact on the test results, enabling HQ quantification in actual water samples.

## 1. Introduction

Hydroquinone (HQ) is an organic compound formed by the substitution of two para-hydrogens of benzene by hydroxyl groups. At room temperature, HQ is a white crystalline or nearly white crystalline powder. At present, HQ is mainly used to produce black-and-white photo-developer compounds, dyes, rubber anti-aging additives, stabilizers, and antioxidants, among others [1]. HQ is highly toxic and accumulates easily in the environment. Due to its high water solubility, it can easily lead to water pollution. In water, HQ can produce a distinctly unpleasant smell even though its concentration is at ppm level and even less than that [2]. Studies have shown that HQ has carcinogenic and mutagenic properties. Symptoms such as headache, tinnitus, nausea, vomiting, abdominal pain, dyspnea, and others can occur after a low-dose ingestion of HQ. In view of its harmful effects, there is a critical need to develop reliable and sensitive analytical methods for the determination of HQ, especially for deployment in environmental pollutant analysis [3]. So far, many analytical methods have been used for HQ quantification, including those based on high-performance liquid chromatography, fluorescence, chemiluminescence, and electrochemical sensing. In that respect, electrochemical methods offer the benefits of elevated sensitivity, accuracy, convenient operation, cheap instrumentation, facile integration, and portability [4]. Moreover, due to the presence of electro-active hydroxyls in its benzene rings, the electrochemical oxidation of HQ is possible [5]. Unfortunately, given the numerous HQ isomers, such as catechol (CC) and resorcinol (RC), common commercial electrodes such as glassy carbon (GCE) and gold electrodes are unable to distinguish their overlapping peaks [6]. Hence, it is necessary to develop new electroactive materials to efficiently detect HQ. In previous literature, researchers mostly used one-dimensional or two-dimensional materials to modify the electrode, which may reduce the catalytic active sites and the specific surface area due to agglomeration.

Given the need for accurate HQ detection and quantification, we propose hydrogel systems carrying active nanomaterials through appropriate interactions [7]. Indeed, hydrogels are increasingly considered for the modification of electrodes due to their high ion diffusion and electron transport, thus improving the sensitivity of electrochemical sensors. Given that typical hydrogels have limited electronic transmission capacity, their modification is required for uses in electrodes and to affect the normal transmission of electrical signals. Thus, given their large specific surface area, as well as their excellent electron and ion transport capacity, we considered conductive systems that combined the electrochemical activity of conductive polymers and the soft properties of hydrogels [8,9].

Polysaccharides are commonly selected to construct hydrogels because of their hydrophilic functional groups which can absorb water and can be readily cross-linked using a variety of chemical and physical agents. Among them, Carboxymethyl chitosan (CMCS) is a biodegradable and biocompatible chitosan (CS) derivative derived from the carboxymethylation of CS. It has numerous advantages such as good water solubility and higher antibacterial activity, as well as moisture retention capacity [10]. Similarly, sodium carboxymethyl cellulose (CMC-Na) is another polysaccharide with good biocompatibility, biodegradability, and low cost [11]. Because of the rich amount of amino and carboxyl groups, CMCS and CMC-Na can interact with other functional materials via physical and chemical techniques, as well as form stable films on solid substrates. In this study, we combined CMCS and CMC-Na, two typical biobased polymers, with a common conductive polymer, polypyrrole. The active amino and hydroxyl groups in biomolecules were effective in the formation of a hydrogel network structure held by hydrogen bonding with the nitrogen atoms in the heterocyclic ring of polypyrrole. Additionally, based on the biobased polymer sodium lignosulfonate (LS), silver nanoparticles were adsorbed with high efficiency and in-situ reduced, and were embedded in the hydrogel network and used to further improve the electro-catalytic efficiency of the system. This silver nanoparticle-embedded conductive hydrogel not only revealed great potential electroanalytical applications for HQ but also developed new avenues for combining the advantage of high permeability to facilitate the diffusion of hydrogels with the high electrical conductivity and catalytic performance of metal ions.

## 2. Materials and Methods

### 2.1. Materials

Ammonium persulfate (APS), sodium hydroxide (NaOH), hydrochloric acid (HCl), trisodium citrate dihydrate (C_6_H_5_Na_3_O_7_·2H_2_O), silver nitrate, pyrrole, sodium acetate (NaAc), acetic acid (HAc), hydroquinone (HQ), and nitric acid (HNO_3_) were all obtained from Sinopharm Chemical Reagent Co., Ltd. (Shanghai, China). Sodium carboxymethyl cellulose (CMC-Na), carboxymethyl chitosan (CMCS), and sodium lignosulfonate (LS) were sourced from Aladdin Reagent (Shanghai, China). Catechol (CC) and resorcinol (RC) were purchased from Macklin regents (Shanghai, China). Tris-HCl was obtained from Beijing Solarbio Technology Co., Ltd. (Beijing, China). Phosphate buffer saline (PBS) was purchased from Hyclone. Ultrapure water used in all experiments was purified by a Milli-Q water purification system (Millipore, Billerica, MA, USA). Pyrrole was distilled as a colorless liquid under the protection of nitrogen before being used.

### 2.2. Methods

Scanning electron microscopy (SEM) was carried out by using a Regulus 8100 system (Hitachi, Tokyo, Japan). Fourier-transform infrared (FT-IR) spectra were obtained with an FT-IR infrared spectrophotometer operated between 400 and 4000 cm^−1^ (Vertex 80V, Bruker, Mannheim, Germany). X-ray diffraction (XRD) patterns were recorded with an Ultima IV diffractometer (Ultima IV, Rigaku, Tokyo, Japan). An electrochemical workstation was used for electrochemical determination (CHI 660e, Chenhua, Shanghai, China).

### 2.3. Hydrogel Preparation

The hydrogel precursor solution was prepared as follows. First, 0.1 g of CMCS, 0.1 g of CMC-Na, and 0.1 g of LS were dissolved in 5 mL of deionized water at room temperature. Then, 200 μL of pyrrole solution was added, followed by stirring at 400 rpm for 5 min until thoroughly mixed. Tris-HCl solution was prepared with 1.21 g of tris-HCl dissolved in 90 mL of deionized water at room temperature. Then, the pH of the solution was adjusted to 7 with 0.1 M HCl or NaOH solution. For HAc-NaAc solution, 1.361 g of NaAc·3H_2_O was dissolved in 90 mL of deionized water. Then, 0.718 mL of HAc was added into the solution. Next, the pH of the solution was adjusted to 7 with 0.1 M HCl or NaOH solution.

### 2.4. Modified Glassy Carbon Electrode (GCE)

Before modification, a GCE was successively polished with 0.3 and 0.05 mm aluminum slurries on the polishing pad and rinsed thoroughly with ultrapure water, yielding a mirror-like surface [12]. Then 0.114 g APS was dissolved in 1 mL ultrapure water to prepare 0.5 mol/L APS aqueous solution. Next, 200 μL APS solution was added into the hydrogel precursor solution and stirred quickly for about 15 s. An amount of 8.0 μL of the mixture solution was carefully dropped on the GCE surface. The hydrogel was formed after ca. 1 min, upon gelation. Soon afterwards, the GCE, which was washed thoroughly, was dipped in 20 mL silver nitrate solution (0.05 M) for about 4 h. Thereafter, the GCE was removed from the silver nitrate solution and rinsed thoroughly with deionized water to remove the silver nitrate attached to the hydrogel surface, and then it was impregnated with 20 mL sodium citrate solution (0.05 M) for another 1 h so that the silver ions inside the gel could be fully reduced to silver nanoparticles (Figure 1). The modified GCE was then rinsed thoroughly with ultrapure water in order to remove excess materials.

### 2.5. Electrochemical Measurement

All electrochemical experiments were carried out in a three-electrode cell. The modified GCE was used as a working electrode and a platinum wire acted as the counter electrode. Saturated calomel electrode was used as a reference electrode. Cyclic Voltammetry (CV) was performed in 0.1 M different buffer solutions (pH = 7) with 10^−4^ M HQ, while electrochemical impedance spectroscopy (EIS) measurements were performed in 0.10 M Tris-HCl buffer with 10^−4^ M HQ, containing 5 mM Fe(CN)_6_^3−^/Fe(CN)_6_^4−^, pH 7. The scan rate for CV measurements was 0.1 V/s, while the frequency for EIS measurements ranged from 0.1 to 10^5^ Hz with 5 mV AC amplitude. Differential pulse voltammetry (DPV) tests were carried out with the pulse amplitude set as 0.05 V, the pulse time set as 0.05 s, and the scan rate set as 0.02 V/s.

### 2.6. Statistical Analysis

The results of CV, EIS, and DPV tests were examined in parallel at least three times. Data were expressed in the following figures with error bars. Error data were provided by Origin Software calculation.

## 3. Results and Discussion

### 3.1. Silver-Embedded Conductive Hydrogel (Ag NP-CH)

SEM, energy spectrum analysis, element mapping, FT-IR spectra, and XRD were performed to confirm the successful synthesis of the silver nanoparticle-embedded conductive hydrogel (Ag NP-CH). The Ag NP-CH showed a loose porous structure with a pore diameter of about 100 μm, as shown in Figure 2A. C, N, O, S, and Ag elements were evenly distributed in the gel. The FT-IR spectrum of the SDCH after freeze-drying revealed a broad absorption band at 3416 cm^−1^, attributed to -OH stretching. The broad absorption bands around 2800 cm^−1^ corresponded to the stretching of C–H, while that at 1071 cm^−1^ corresponded to the C–O–C stretching vibration of CMCS and CMC-Na. The FT-IR spectra of polypyrrole (Figure 2B) included peaks at 1583 cm^−1^ and 1387 cm^−1^ assigned to the C–C and the C–N vibrations of the pyrrole ring, respectively. In addition, peaks at 1261, 1071, and 905 cm^−1^ corresponded to the characteristic –CH–H plane vibration, N–C strain vibration, C–H deformation vibration, and in-plane C–C vibrations of the polypyrrole structure, respectively [13,14,15].

Figure 2C shows the XRD pattern of Ag NP-CH that presented three characteristic diffraction peaks at 38.3, 44.3, and 64.5 in the 2θ region, which corresponded to the (111), (200), and (220) planes of Ag [16]. In addition, the broad peaks between 25 and 35 were attributed to the short-range-ordered structure of polypyrrole [17]. All in all, the results from the chemical and structural analyses indicated the successful synthesis of Ag NP-CH.

### 3.2. Electrochemical Activity

Cyclic Voltammetry (CV) and electrochemical impedance spectroscopy (EIS) were effective to study the assembly of the modified electrode [18]. All the studies were performed in 0.10 M Tris-HCl buffer with 10^−4^ M HQ, containing 5 mM Fe(CN)6^3−^/Fe(CN)6^4−^, pH 7. Compared with the CV profile of the bare electrode before modification, the conductive hydrogels (CH) attached to the GCE surface produced a significantly enhanced electrochemical response signal. Further reduction of silver in the gel produced a redox peak at −0.4 V (Figure 3A) [19]. The Nyquist plot shown in Figure 3B agreed with the CV curves shown in Figure 3A. The diameter of the semicircle at higher frequencies corresponded to the electron transfer resistance (Rct) between the surface of the electrode and [Fe(CN)6]^3−/4−^, while the linear part at a lower frequency corresponded to the diffusion process. The inset of Figure 3B exhibits the equivalent circuit of the electrochemical system which involved charge-transfer (Rct), Warburg impedance (Zw), solution resistance (Rs), and constant phase element (CPE). The Rct of bare GCE, CH/GCE (black line), and Ag NP-CH/GCE (blue line) were calculated to be 1448.5 Ω, 1174.6 Ω, and 751.2 Ω, respectively. Compared with the Rct of bare GCE, the Rct of CH/GCE (black line) and Ag NP-CH/GCE (blue line) was slightly lower, indicating that the construction of gels on the surface of GCE favored electron transfer [20]. Additionally, these results clearly demonstrate that Ag NPs hosted in the chitosan hydrogel increased the catalytic activity of the GCE, improving the kinetics and the thermodynamics of the electrochemical process involved in the HQ electroanalysis [21].

### 3.3. Optimization of the Experimental Conditions

Detection optimization was carried out for suitable electrochemical determination and sensitivity. Here, the selection of electrolytes and the regulation of pH were optimized. Figure 3C,D shows the redox current peaks obtained on the CV curves under different electrolyte solutions and pH values, respectively. It can be observed that compared with the other two electrolyte solutions, the sensor produced a better current intensity response in the Tris-HCl solution. The current intensity response reached a max value at pH = 7. We speculated that this phenomenon is due to the structure of HQ itself. HQ is a protic aromatic molecule (pKa = 9.852). Under high pH conditions, HQ can easily undergo deprotonation and turn to anions; while at low pH values, it is protonated in the form of –OH_2_^+^. Therefore, under acidic conditions, the gradual deprotonation of HQ leads to the increase of their oxidation currents. It is worth noting that when the pH value is greater than 7, the increase of hydroxyl ions in the solution may reduce the adsorption capacity of HQ on the electrode surface, resulting in a decrease in the oxidation currents. Therefore, we carried out tests in Tris-HCl solution at pH = 7 for further electrochemical determination.

For the purpose of explaining the electro-oxidation process of HQ on the surface of Ag NP-CH/GCE, the CV response curves corresponding to 100 μM HQ in 0.1 M Tris-HCl were recorded (scanning rate of 10 to 500 mV/s). As shown in Figure 3E, the redox peak of HQ appeared at 0.136/−0.057 V at a scan rate of 10 mV/s. The potential difference between anodic and cathodic peaks was 0.193 V. The potential difference increased with the increased scan rate. This fact indicated that as the scan rate increased, the redox reaction became irreversible at Ag NP-CH/GCE. Upon data analysis, it was found that the redox peak current intensities increased as the scan rate increased, in the range from 10 to 500 mV/s. The linear relationship between the peak current intensity and the square root of the scan rate was Ipa(μA) = 76.72 × v(V/s)^1/2^ + 8.70 (R^2^ = 0.97) and Ipc(μA) = −37.45 × v(V/s)^1/2^ −6.02 (R^2^ = 0.96) (Figure 3F). This result implies that the oxidation and reduction reaction of HQ that occurred on the surface of Ag NP-CH/GCE was diffusion-controlled [22,23].

### 3.4. Electrochemical HQ Sensing

Under optimal experimental conditions, DPV tests were carried out to record the current intensity response at varying HQ concentrations. As depicted in Figure 4A, with the increased HQ concentration from 0.1 to 100 μM, the current intensity values increased. The obtained calibration plot from triplicate experiments is shown in Figure 4B. Specifically, the linear regression equation corresponded to the following: Ipa (μA) = 0.62 CHQ (μM) + 2.45 (R^2^ = 0.99). Additionally, based on the equation, the detection limit was determined to be 3 × Sb/As (Sb is the blank standard deviation and As is the absolute value of slope). Hence, the detection limit was calculated to be 0.12 μM (Appendix A). The proposed sensor compares well with other HQ sensors reported in the literature, especially considering the composition of the systems, as shown in Table 1.

### 3.5. Selectivity, Reproducibility, and Stability of Ag NP-CH/GCE

Interference substances coexist with the target analyte, affecting the test results [39]. Here, CC and RC, the isomers of HQ, were respectively added into the solution to evaluate the selectivity of Ag NP-CH/GCE. The anodic peak potentials of the isomers from HQ, CC, and RS were 0.11, 0.23, and 0.64 V, respectively (Figure 5A). This is because different relative positions of hydroxyl groups have different charge densities. The two hydroxyl groups of HQ are in the para position, which makes for the highest charge density. The larger charge density makes the molecule easier to oxidize. Hence, the oxidation potential of HQ was the lowest. Given this fact, the addition of 30 μM CC or RS does not have a significant impact on the test results. At this latter concentration, the peak current intensity was stable at 22 A. At the same time, we also tested the interference of inorganic ions in the determination. Here, we added 1 mM of different inorganic ions to the HQ test solution. As shown in Figure 5B, due to the fact that different ions have different REDOX potentials, the addition of inorganic ions also has no obvious influence on the determination results.

Reproducibility and stability are suitable indicators of sensor performance [40]. Here, to verify the reproducibility of Ag NP-CH/GCE, eight different electrodes were fabricated and tested in a solution containing 30 μM HQ. The relative standard deviation of the anodic peak current intensity was 1.37%, indicating a good reproducibility of Ag NP-CH/GCE (Figure 5C and Appendix A). In addition, to assess the stability of Ag NP-CH/GCE, the electrochemical performance of the electrode was tested after one week of storage in 0.1 M Tris-HCl buffer solution at 4 °C. After eight consecutive tests on alternating days, the anodic peak current intensity was 93.4% of the initial current intensity, indicating a good stability of the sensor (Figure 5D and Appendix A).

### 3.6. Practical Sample Analysis

In order to evaluate the on-site monitoring and performance of the presented system, actual water samples were chosen for analysis (tap water was collected at Nanjing Forestry University, and river water was collected from the small pond on campus). Firstly, a series of the water samples were diluted 10-fold with 0.1 M Tris-HCl buffer solution, then different concentrations of HQ were spiked to a given sample volume. DPV measurements were applied to calculate the recoveries of HQ. Table 2 presents the recovery efficiency of HQ in the samples. The recovery rates were between 98.4% and 103.9%, which demonstrated the excellent potential for HQ determination in practical samples.

## 4. Conclusions

We successfully developed a conductive hydrogel based on carboxymethyl chitosan, sodium carboxymethyl cellulose, and polypyrrole via hydrogen bonding. By loading the hydrogel with sodium lignosulfonate, used for in-situ reduction of silver ions, silver nanoparticles were incorporated into the network, which improved the electro-catalytic efficiency. The system was designed for a high hydrogel permeability to facilitate diffusion and electron transport of analytes. The synthesized sensor exhibited high selectivity and sensitivity towards HQ. The oxidation current density peak of HQ was linear over the 0.1–100 μM concentration range, with a detection limit as low as 0.12 μM (signal-to-noise of 3). In addition, this sensor possessed high reproducibility and stability, with no interference. The proposed system was shown to perform suitably for the analysis of actual water samples. In summary, this silver nanoparticle-embedded conductive hydrogel not only revealed great potential electroanalytical applications for HQ but also developed new avenues for combining the advantages of the high permeability to facilitate diffusion of hydrogels with the high electrical conductivity and catalytic performance of metal ions. Still, there are some places worth perfecting and breaking through for our proposed electrochemical, such as further design to improve the efficiency of electrocatalysis, thus improving the sensitivity, and using a stronger force to fix the gel on the surface of GCE in order to avoid falling off after multiple tests. We will make great efforts to further overcome these technical limitations in future work.

## Figures and Tables

**Figure 1 polymers-15-02424-f001:**
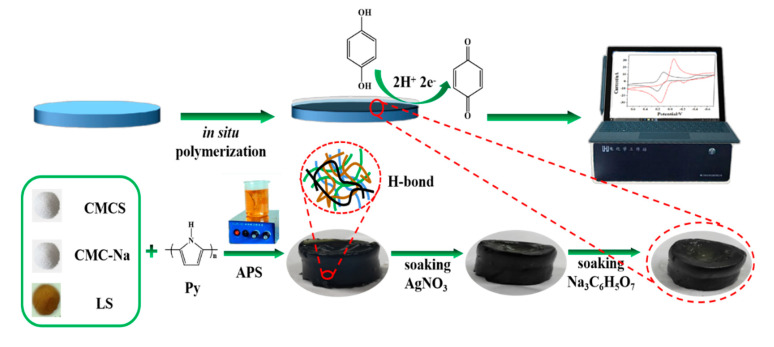
Fabrication of silver nanoparticle-embedded conductive hydrogel (Ag NP-CH) on a glassy carbon electrode (GCE).

**Figure 2 polymers-15-02424-f002:**
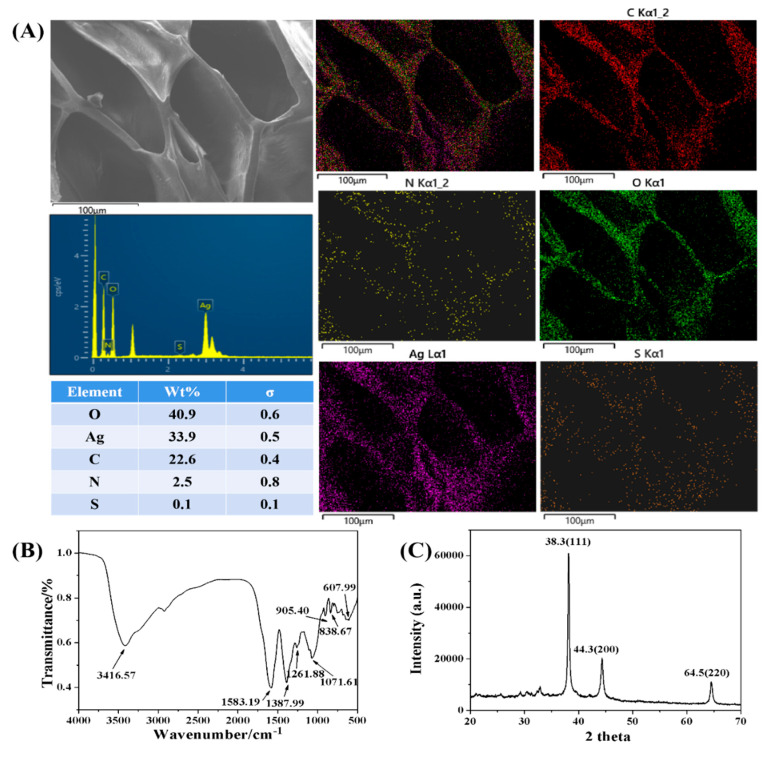
(**A**) SEM image and corresponding element mapping and energy spectrum of the surface of Ag NP-CH/GCE; (**B**) FT-IR; and (**C**) XRD spectra of Ag NP-CH.

**Figure 3 polymers-15-02424-f003:**
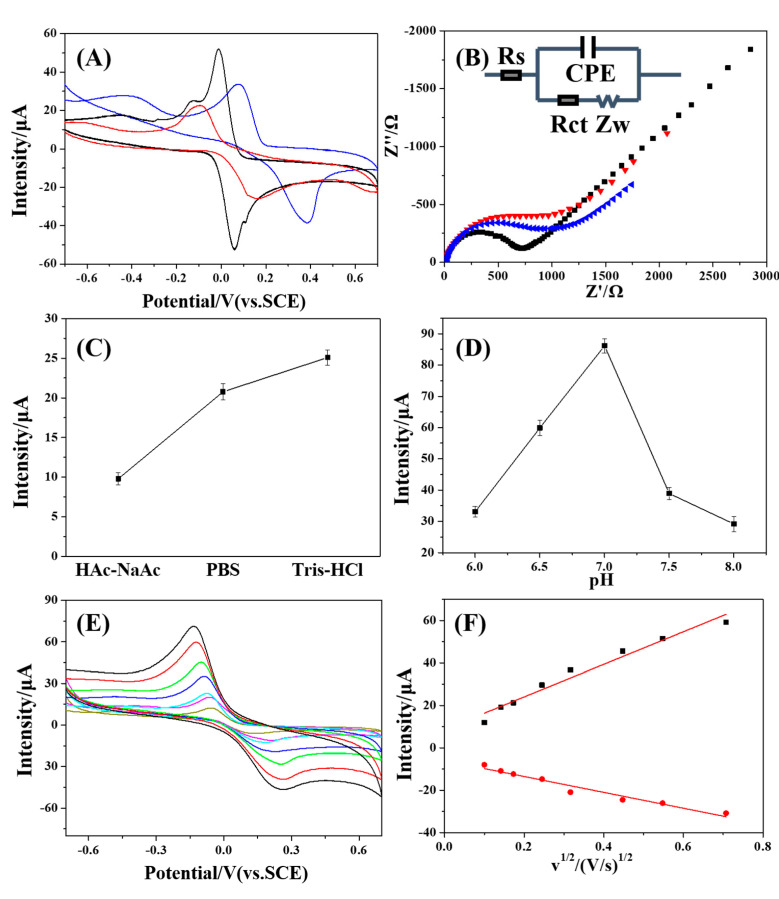
(**A**) CVs of 1.0 × 10^−4^ mol/L HQ at bare GCE (red line), CH (black line), and Ag NP-CH/GCE (blue line) in 0.10 M Tris-HCl buffer, pH 7, scan rate: 0.1 V/s. (**B**) EIS of 1.0 × 10^−4^ mol/L HQ at bare GCE (red line), CH (black line), and Ag NP-CH/GCE (blue line) in 0.10 M Tris-HCl buffer, containing 5 mM Fe(CN)_6_^3−^/Fe(CN)_6_^4−^, pH 7. (**C**) Anodic peak current intensities for different buffer solutions (pH = 7, 1.0 × 10^−4^ mol/L HQ). (**D**) Anodic peak current intensities at different pH values (0.10 M Tris-HCl buffer, 1.0 × 10^−4^ mol/L HQ). (**E**) CV spectra of 1.0 × 10^−4^ M in 0.10 M Tris-HCl (pH = 7.0) at different scan rates. The signals shown, going from the inner to the outer spectra, corresponded to 0.01, 0.02, 0.03, 0.06, 0.1, 0.2, 0.3, and 0.5 V/s. (**F**) Linear fit to the peak current intensities as a function of the square root of scan rate of HQ.

**Figure 4 polymers-15-02424-f004:**
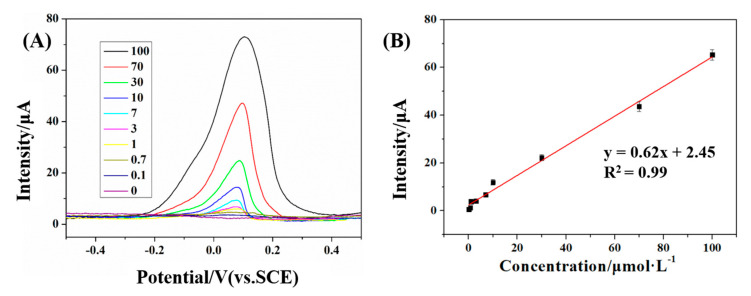
(**A**) DPVs of Ag NP-CH/GCE at various HQ concentrations = (0, 0.1, 0.7, 1, 3, 7, 10, 30, 70, and 100 μM). (**B**) Calibration plot of HQ (pulse amplitude: 0.05 V; pulse width: 0.05 s; scan rate: 0.02 V/s; electrolyte solution: Tris-HCl solution at pH = 7; each point was measured three times).

**Figure 5 polymers-15-02424-f005:**
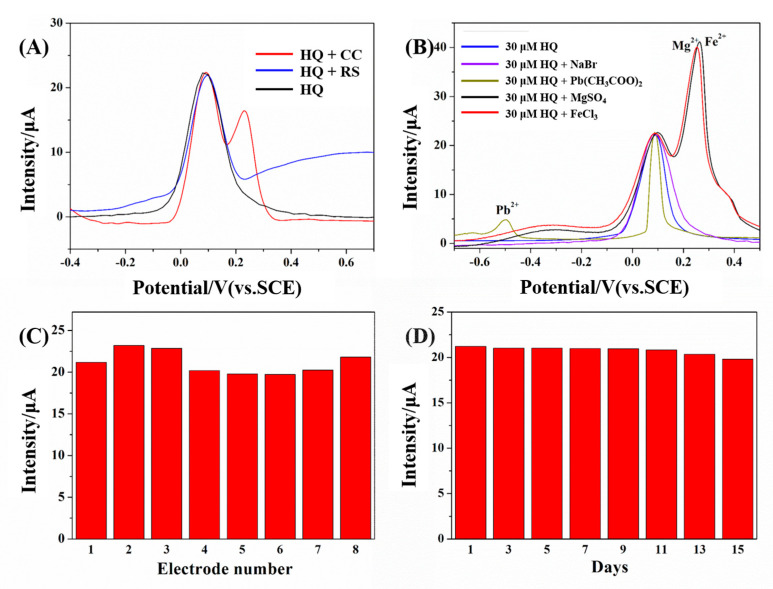
(**A**) DPVs of Ag NP-CH/GCE with 30 μM HQ, 30 μM HQ and 30 μM CC, and 30 μM HQ and 30 μM RS, respectively. (**B**) DPVs of Ag NP-CH/GCE with 30 μM HQ with 1 mM of different kinds of inorganic ions. (**C**) Current intensity response for eight different electrodes with 30 μM HQ and (**D**) current response obtained on one electrode that was kept in Tris-HCl buffer solution at 4 °C for one week and detected every other day with 30 μM HQ.

**Table 1 polymers-15-02424-t001:** Comparison of analytical performance (linear range and limit of detection) of Ag NP-CH/GCE with previously reported electrochemical sensors for the determination of HQ.

Electrochemical Sensor	Linear Range (μM)	LOD (μM)	Ref.
Fe_2_O_3_/CNTs/FTO electrode	1.0~80.0	0.5	[24]
P-L-Cys/Au_1.5_Pt_1_Co_1_/GCE	0.1~200	0.045	[25]
MIL-101(Cr)-rGO-2-CPE	4~1000	0.66	[26]
COF-3-BPPF6-CPE	2~2000	0.31	[27]
ZIF-8/CNF/GCE	2~510	0.06	[28]
NSC/CPE	0.01~700	0.0103	[29]
NPG electrode	0.2~100	0.083	[30]
CuO/GCE	0.3~250	0.009	[31]
Pt/Poly(Isoleucine)/GCE	0.1~100	0.08	[32]
Au-gC_3_N_4_-MOF-CPE	0.005~100	0.001	[33]
graphene/Ir(III) complex/GCE	0.05~100	0.001	[34]
MgOMPCPE	10~100	0.11	[35]
alk-Ti_3_C_2_/N-PC/GCE	0.5~50	0.0048	[36]
MgO/GO/ MCPE	50~400	0.3	[37]
Co_3_O_4_@carbon-_2_/GCE	0.8~127.1	0.03	[38]
**Ag NP-CH/GCE**	**0.1–100**	**0.12**	**This work**

**Table 2 polymers-15-02424-t002:** Determination of HQ in real samples.

Sample	Added (μM)	Measured (μM)	Recovery (%)	RSD (%)
Tap water	0	0	-	-
	10	10.4	103.9	2.34
	30	31.0	103.5	3.94
	50	49.2	98.4	2.88
River water	0	0	-	-
	10	9.9	98.7	3.47
	30	30.9	103.3	2.36
	50	51.3	102.7	4.53

## Data Availability

The data presented in this study are available on request from the corresponding authors.

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
