# Peer review of "Silver Nanoparticle-Embedded Conductive Hydrogels for Electrochemical Sensing of Hydroquinone"

_polymers, 2023, doi:10.3390/polym15112424_

Round 1
Reviewer 1 Report
This study is interesting and within the scope of the journal; however, a few comments appended below should be considered as follows:
1. Abstract: the authors should add quantitative data to strengthen the abstract.
2. Introduction: The authors should elaborate on the importance of carboxymethyl chitosan and sodium carboxymethyl cellulose in several applications. They also should highlight the novelty of this study. Some pertinent reports could be incorporated in this section;
3. Methodology: please add a section for statistical analysis at the end, indicating the number of replicates, test and software used in this study.
4. Results and discussion: It should be improved by comparing your study with previous reports. All the data in figures should be included as mean ± SD with statistical difference.
Author Response
Thank you for your comments concerning our manuscript entitled “Silver nanoparticles-embedded conductive hydrogels for elec-trochemical sensing of hydroquinone” (ID: polymers-2328326). Those comments are all valuable and very helpful for revising and improving our paper, as well as the important guiding significance to our research. We have studied comments carefully and have made correction which we hope meet with approval. Every revised portion is marked with red color in the paper.

Reviewer 2 Report
In summary, the researchers developed a conductive hydrogel using carboxymethyl chitosan and sodium carboxymethyl cellulose that was able to effectively couple with polypyrrole using hydrogen bonding. By loading the hydrogel with sodium lignosulfonate, silver nanoparticles were incorporated in the network, leading to an improvement in electro-catalytic efficiency. The hydrogel precursor was cast onto glassy carbon electrodes to create an electrochemical sensor with high sensitivity and selectivity towards HQ. Overall, the developed sensor has the potential to be used for various applications where highly sensitive and selective detection of HQ is required.
1.In the introduction, there are many conductive hydrogels. The authors should mention them, like, in some papers, they are using the RU compounds, which show the conductivity, Contraction waves in self-oscillating polymer gels. I can not see the originality of this paper, and why they use Silver nanoparticles. The introduction should be re-constructed.
2. Fabrication of silver nanoparticles-embedded conductive hydrogel (SDCH) on a glassy carbon electrode (GCE). In this process, their fabrication parameters should be classified, like stirring speeds or other environmental parameters.
3. They obtained excellent results in figure.3. is it possible to make a flexible sensor or others?
4. This paper lacks some theories.
Author Response

(The authors gave the same response as above.)

Reviewer 3 Report
In this research, an electrochemical sensor has been developed for the determination of hydroquinone by modifying a glassy carbon electrode with silver nanoparticle-contained chitosan hydrogel. The conductive hydrogel was synthesized using carboxymethyl chitosan, carboxymethyl cellulose, polypyrrole, and lignosulfonate.
The study is interesting, but the manuscript is incomplete. It remains to scientifically discuss the results obtained. Therefore, I recommend that a minor review be performed before the manuscript can be considered for publication in Polymers.
General comments:
1. The acronyms used by the authors to indicate hydrogels and modified electrodes need to be revised. It's very confusing. Nanoparticulate silver is usually presented as Ag NPs in the literature. Therefore, I recommend using Ag NP-CH, and in the modified electrode Ag-NP-CH/GCE. GCE must appear on all modified electrodes.
2. Electrode preparation is very time-consuming. Hours to prepare an electrode. So, how many measurements are possible with a modified electrode, that is, without the need for mechanically cleaning the surface and preparing a new film?
3. Update the terminologies for the electrochemical methods according to new IUPAC recommendations. See the information in Pure and Applied Chemistry, 92 (2020) 641–694. Current is I (in italics).
4. The authors, like many others, confuse the terms "detection" and "determination". Detection is qualitative by nature, while determination always is quantitative. Qualitative analysis is the detection of the presence of ions or compounds in an unknown sample, for example. The term "determination" refers to quantitative analysis to obtain data on the amount of analyte by weight or by the concentration of an element or a compound in a sample. Therefore, most of the words “detection" in the manuscript should be replaced by the term "determination" (or "quantitation" or "assay") if quantitative assays are involved.
Specific comments:
1. Introduction:
a. Lines 43-44. It is important to add the advantages of using electrochemical methods. So, add the sentence: “In that respect, electrochemical methods offer the benefits of elevated sensitivity, accuracy, convenient operation, cheap instrumentation, facile integration, and portability”. Add the reference Chemosensors, 10 (2022) 357 to validate this information.
b. Line 47. The correct is “glassy carbon”.
c. Line 61. An important advantage of chitosan is the ability to form stable films on substrates (electrodes). Thus, add the sentence: " Furthermore, chitosan has the ability to form stable films on solid substrates."
2. Experimental:
a. Line 76. Check. HNO3.
b. Line 78. Enter the molecular weight of chitosan (low, medium or high).
c. Line 85. Separate the hydrogel preparation into a new subsection.
d. Line 90. Check. H2O.
e. Line 101. Inform the geometric area of GCE.
f. Line 126. DPV parameters are pulse amplitude, scan rate, and pulse time. Check.
g. You need to create a section to describe sample preparation. Describe the samples, where they were acquired (company and city). How they were prepared for the analyses.
3. Results and discussion:
a. Line 154. It needs to be clarified that the study was made for HQ.
b. Line 155. The superior result provided by the presence of Ag NPs in the electrode needs to be explained. Add the following sentence: These results clearly demonstrate that Ag NPs hosted in the chitosan hydrogel increased the catalytic activity of the GCE, improving the kinetics and the thermodynamics of the electrochemical process involved in the HQ electroanalysis. Add the reference Analytical and Bioanalytical Chemistry, 408 (2016) 2595–2606 to validate this information.
c. Line 158. Check.
d. Line 172. Why was the best response obtained at pH 7.0? What is the pKa value of HQ? Enhance the discussion.
e. Figure 3A. What were the HQ peak-to-peak separation values? Discuss the reversibility of the HQ electrochemical system in different electrode modifications.
f. Figure 4. The figure caption needs to have as much information as possible. Add DPV parameters, buffer solution, pH, and number of replicates.
g. Indicate on the x-axis of the voltammograms which reference electrode was used.
h. Table 2. Indicate that without spiked (0) the presence of HQ was not detected in the water samples.
i. Line 247. The recovery assay provides the accuracy of the method.
4. Conclusion:
a. The conclusion is too long and similar to the abstract. Enhance the text of the conclusion. The conclusion is not abstract.
Author Response

(The authors gave the same response as above.)

Round 2
Reviewer 2 Report
the authors improved the manuscript significantly.
Reviewer 3 Report
I recommend that the manuscript be accepted.